# Hemophagocytic Lymphohistiocytosis Following BNT162b2 mRNA COVID-19 Vaccination

**DOI:** 10.3390/vaccines10040573

**Published:** 2022-04-08

**Authors:** Ting-Yu Lin, Yun-Hsuan Yeh, Li-Wen Chen, Chao-Neng Cheng, Chen Chang, Jun-Neng Roan, Ching-Fen Shen

**Affiliations:** 1Department of Pediatrics, National Cheng Kung University Hospital, College of Medicine, National Cheng Kung University, Tainan City 704, Taiwan; n102360@mail.hosp.ncku.edu.tw (T.-Y.L.); i54981328@gs.ncku.edu.tw (Y.-H.Y.); z107a0034@email.ncku.edu.tw (L.-W.C.); can66@mail.ncku.edu.tw (C.-N.C.); 2Department of Pathology, National Cheng Kung University Hospital, College of Medicine, National Cheng Kung University, Tainan City 704, Taiwan; n042707@mail.hosp.ncku.edu.tw; 3Institute of Clinical Medicine, College of Medicine, National Cheng Kung University, Tainan City 704, Taiwan; 102h0019@gs.ncku.edu.tw; 4Medical Device Innovation Center, National Cheng Kung University Hospital, College of Medicine, National Cheng Kung University, Tainan City 704, Taiwan; 5Division of Cardiovascular Surgery, Department of Surgery, National Cheng Kung University Hospital, College of Medicine, National Cheng Kung University, Tainan City 704, Taiwan

**Keywords:** mRNA COVID-19 vaccine, hemophagocytic lymphohistiocytosis, hyperinflammatory syndrome

## Abstract

Although serious adverse events have remained uncommon, cases of myocarditis induced by messenger RNA (mRNA) COVID-19 vaccines have been reported. Here, we presented a rare but potentially fatal disorder, hemophagocytic lymphohistiocytosis, in a 14-year-old previously healthy adolescent after BNT162b2 mRNA vaccination. The initial evaluation showed splenomegaly, pancytopenia, hyperferritinemia, and hypofibrinogenemia. Further examination revealed positive blood EBV DNA, and other infectious pathogen surveys were all negative. Hemophagocytosis was observed in the bone marrow aspiration and biopsy. HLH was confirmed and intravenous immunoglobulin (IVIG) and methylprednisolone pulse therapy were given. Venoarterial extracorporeal membrane oxygenation (VA-ECMO) was set up for cardiopulmonary support for 3 days due to profound hypotension. The patient was kept on oral prednisolone treatment for 28 days with the following gradual tapering. The hemogram and inflammatory biomarkers gradually returned to normal, and the patient was discharged. The fulminant presentation of HLH in our case could be the net result of both acute immunostimulation after COVID-19 vaccination and EBV infection. Our case suggests that the immune activation after COVID-19 vaccination is likely to interfere with the adequate immune response to certain infectious pathogens, resulting in a hyperinflammatory syndrome.

## 1. Introduction

Messenger RNA (mRNA) vaccines have been widely used against severe acute respiratory syndrome coronavirus 2 (SARS-CoV-2) infection with excellent efficacy [1]. However, post-marketing surveillance discovered the BNT162b2 mRNA COVID-19 vaccine is associated with an excessive risk of myocarditis [2]. Herein, we present another rare but potentially fatal disorder, hemophagocytic lymphohistiocytosis (HLH), in a previously healthy adolescent after BNT162b2 vaccination.

## 2. Case Presentation

This 14-year-old girl presented to a regional hospital on October 15 (day 0), 2021, with 4 days of fever, headache, nausea, and progressive tachypnea. Before this, she received the first dose of the BNT162b2 vaccine on September 30 (day −15). At the emergency department, she had drowsy consciousness, mottling skin, jaundice, and hypotension. The laboratory data revealed leukopenia (white cell count (WBC), 2000/μL), thrombocytopenia (platelet count, 44 × 10^3^/μL), elevated C-reactive protein (CRP, 88 mg/L) and ferritin (4254.2 ng/mL), direct hyperbilirubinemia (total/direct bilirubin, 7.62/5.31 mg/dL), and liver function impairment (aspartate aminotransferase, 142 U/L; alanine aminotransferase, 182 U/L). The cardiac enzyme was normal on the first blood test. Abdominal computed tomography demonstrated splenomegaly and enlarged lymph nodes at the hepatic hilum. The nasal swab for the SARS-CoV-2 polymerase chain reaction (PCR) was negative. She soon developed pulmonary hemorrhage, acute respiratory distress syndrome, hypotension and was intubated for respiratory support after hospitalization. Meanwhile, progressive pancytopenia (WBC, 1200/μL; hemoglobin, 8.6 g/dL; platelet count, 45 × 10^3^/μL), hyperferritinemia (>40,000 ng/mL), hypofibrinogenemia (1.43 g/L), severe lactic acidosis (11.3 mmol/L), disseminated intravascular coagulopathy, and elevated cardiac enzyme (troponin-I, 2982.2 pg/mL; NT-proBNP > 70,000 pg/mL) were noted on day 2. Hemophagocytic lymphohistiocytosis (HLH) was confirmed based on the HLH-2004 diagnostic criteria (fulfilling five out of the eight criteria: fever, splenomegaly, cytopenias affecting at least two of three lineages, hypofibrinogenemia ≤1.5 g/L, and hyperferritinemia). This patient received one dose of intravenous immunoglobulin (IVIG) of 1 g/kg and was then transferred to our hospital on day 2 (Figure 1).

After hospitalization, bone marrow aspiration revealed significant hemophagocytosis (Figure 2A), and the histopathology examination demonstrated both hemophagocytosis and Epstein–Barr virus-encoded small RNA (ribonucleic acids) (EBER)-positive lymphocytes by in situ hybridization, confirming the diagnosis of Epstein–Barr virus (EBV)-associated HLH (Figure 2B–D). Methylprednisolone pulse therapy (1000 mg/day) and the second dose of IVIG were given. Owing to profound hypotension despite high inotropic agents use, venoarterial extracorporeal membrane oxygenation (VA-ECMO) was set up for cardiopulmonary support on day 2, withdrawn after the condition stabilized on day 5, with extubation on day 10. Because of decreased spontaneous movement, computed tomography (CT) of the brain was arranged on day 4 and showed intracranial hemorrhage at right parietal and bilateral occipital lobes and subarachnoid hemorrhage at bilateral high frontal–parietal and parafalcine regions. Following 5 days of methylprednisolone pulse therapy, she was kept on oral prednisolone (40 mg/m^2^) treatment for 28 days, followed by gradual tapering and discontinuation on day 43. The hemogram and the inflammatory biomarkers gradually returned to normal without chemotherapeutic medications (Figure 1A,B). This patient was then released from hospital on day 28.

The blood EBV DNA was detectable at a high level in the acute stage of the illness (1,980,000 IU/mL on day 2; 738,000 IU/mL on day 3, down to 146 IU/mL on day 8), and became undetectable on day 17. Bone marrow examination was performed on day 3, but we did not send the EBV DNA from bone marrow cells due to the inadequate specimen. Follow-up bone marrow EBV on day 12 was 103 IU/mL and became undetectable on day 31. The saliva EBV DNA was not measured. We used the Abbott RealTime EBV assay to perform an EBV DNA quantitative amplification test, and its detected limit is 1.60 log_10_ IU/mL (40 IU/mL/108 copies/mL) for plasma specimens. There was a fourfold decrease in the EBV IgG titer in serial sera (with a titer of 1:320 (+) on day 2, 1:160 (+) on day 9, and 1:80 (+) on day 17, and remained 1:80 (+) on day 31). The serial EBV IgM were all negative, and the EBNA-1 IgG were negative on day 0. The serum interleukin 6 (IL-6) level was 1151 pg/mL on day 0, with the peak at 26,963 pg/mL on day 1, and then gradually decreased (Figure 1B). The serum IL-6 level reached the peak one day earlier than ferritin and 3 days earlier than CRP, and also declined earlier than ferritin and CRP. Other pathogen surveys included human herpesvirus 6, parvovirus B19, cytomegalovirus, herpes simplex virus, hepatitis A, B, and C, human immunodeficiency virus (HIV), SARS-CoV-2 PCR, and multiplex PCR using the BioFire FilmArray respiratory panel (BioFire Diagnostics, Salt Lake City, UT, USA). The FilmArray respiratory panel was performed by mixing 300 μL of the viral transport medium with the sample buffer, injection into a test pouch containing reagents for nucleic extraction, PCR amplification, and detection of pathogen targets, including human adenovirus, human coronavirus (HCoV)-229E, HCoV-HKU1, HCoV-OC43, HCoV-NL63, HMPV, human rhinovirus/enterovirus, influenza A (A, A/H1, A/H1-2009, A/H3), influenza B, human parainfluenza viruses 1–4, respiratory syncytial virus, and three bacteria (*Bordetella pertussis*, *Chlamydia pneumoniae*, and *Mycoplasma pneumoniae*). All these pathogen surveys except EBV were negative. We followed up on the blood EBV DNA viral load 1 month and 3 months after discharge, and the results were all undetectable. We also checked the hemogram and inflammatory biomarkers at 1 month and 3 months after discharge, and the results remained normal.

## 3. Discussion

Hemophagocytic lymphohistiocytosis (HLH) is a rare but potentially life-threatening disorder. It is manifested as a severe hyperinflammatory syndrome induced by aberrantly activated macrophages and cytotoxic T cells. HLH could happen without a pre-existing medical condition, or secondary to a malignant, infectious, or autoimmune/autoinflammatory stimulus [3]. An HLH patient is often critically ill with progressive multiple organ failure and requires intensive care. If untreated, the mortality could be 50%, as reported for presumably acquired cases in children [4].

We report here the first case of EBV-associated HLH in a previously healthy adolescent following BNT162b2 vaccination. ECMO was employed to provide temporary hemodynamic support, allowing her to recover the cardiac and pulmonary function compromised during the initial cytokine storm. To categorize the EBV infection status, we used commercial immunoassays to detect viral capsid antigen (VCA) IgG, VCA IgM, and EBV nuclear antigen (EBNA)-1 IgG [5]. In the primary EBV infection, VCA IgM was only produced transiently during the early phase of the illness, while VCA IgG, once generated, could persist for a lifetime. As for EBNA-1 IgG, it is not observed in the acute phase of EBV infection but slowly appears two to four months after the onset of symptoms and persists for the rest of a person’s life. Theoretically, the presence of VCA IgM and VCA IgG without EBNA-1 IgG indicates acute infection, whereas the presence of VCA IgG and EBNA-1 IgG without VCA IgM is typical of a past infection [6]. However, in a clinical scenario, VCA IgM are not necessarily produced in all patients with the primary infection, and also not everyone would generate EBNA-1 IgG antibodies. G. Bauer reported that there the anti-EBNA-1 response is missing in 5% of healthy individuals, and loss of anti-EBNA-1 may be observed during immunosuppression even some months and sometimes years after the primary infection [7]. In our patient, the EBV IgG was positive with a titer of 1:320 (+) on day 2, 1: 160 (+) on day 9, and 1:80 (+) on day 17. The EBV IgM and EBNA-1 IgG were negative. Both acute or past infection would be possible according to this serological profile. Repeating the serology test after a reasonable period of time in order to detect any changes in the antibody profile might be helpful to provide additional information. Furthermore, VCA IgG avidity, which is of low levels in the course of a recent infection and of high levels in the course of a past infection or reactivation, could be useful [8,9]. Secondly, anti-early antigen (EA) IgG are usually expressed during the early phase of viral replication and generally fall to undetectable levels after three to six months. In many people, detection of the antibody to the EA is a sign of active infection. However, both VCA IgG avidity and EA IgG are not available in our institute.

In our case, we used the Abbott RealTime EBV assay, which targets the EBV BLLF gene, encoding the gp350/220 envelope glycoprotein to measure the viral load. The results were interpreted as follows: target not detected, target detected but not quantifiable (<1.60 log_10_ IU/mL or 40 IU/mL), target detected and quantifiable (1.60 log_10_ IU/mL to 8.30 log_10_ IU/mL), and target detected above the upper limit of quantification (>8.30 log_10_ IU/mL or >200,000,000 IU/mL) for plasma specimens. In a previous report, a 22-year-old female had an extremely high EBV viral load of 23,000,000 copies/mL when diagnosed with HLH. Her EBV viral load reduced to 660 within nine days and her blood counts and liver function returned to normal after treatment with interleukin 1 inhibitor anakinra, methylprednisolone, IVIG, and a single dose of rituximab, suggesting the importance of restoration of the normal immune function along with viral clearance in disease recovery [10].

For the HLH treatment, we gave two doses of IVIG and methylprednisolone pulse therapy for 5 days, followed by oral prednisolone and gradual tapering. We discontinued oral prednisolone on day 43. We followed up on the blood EBV DNA viral load 1 month and 3 months after discharge, and the results were all undetectable. We also checked the hemogram and inflammatory biomarkers at 1 month and 3 months after discharge, and the results remained normal.

Familial hemophagocytic lymphohistiocytosis (FHL) is generally diagnosed with degranulation assays based on surface upregulation of CD107a on stimulated natural killer (NK) cells and cytotoxic T lymphocytes (CTLs), flow cytometry for cell surface expression of perforin, or genetic analysis for FHL gene mutation [11,12]. These functional or molecular examinations are not available in our hospital. Therefore, we did not perform diagnostic studies. There are some sporadic reports of adolescent- and adult-onset FHL, and we cannot totally exclude our case from FHL. However, most FHL cases exhibit a more catastrophically deteriorated course and could rapidly progress to death within weeks without prompt therapy. Even with an aggressive regimen combining corticosteroids, cyclosporine, etoposide, and anti-thymocyte globulins, these cases usually achieve disease control transiently and might encounter a recurrent relapse of the disease during the rest of the lifetime. In such cases, allogeneic hematopoietic stem cell transplantation seems to be the optimal resolution to cure the disease. As for our case, fulminant presentation was noted in the beginning, but the disease was gradually under control only with IVIG and methylprednisolone pulse therapy. The patient’s vital signs soon stabilized, and ECMO was withdrawn three days after. So far, there is no evidence of disease recurrence four months after discontinuing all medication. Although some congenital immunodeficiencies, including Chediak–Higashi syndrome, Griscelli syndrome type 2 (GS2), Hermansky–Pudlak syndrome (HPS), are associated with a high risk of developing HLH, our patient did not present any symptoms of disease mentioned above [12].

Some researches demonstrated that the BNT162b2 vaccine is a plausible trigger for the hyperinflammatory syndrome, explained by the inflammatory generating potential of spike proteins [13], the innate and adaptive immune mechanisms, including molecular mimicry and the potential immune response mediated by anti-spike antibodies [14]. It is known that T cell activation plays a vital role in HLH pathogenesis, and this could be triggered through infection or other antigen stimulation, such as measles and influenza vaccination [15,16]. Otagiri et al. reported a 19-month-old girl developed HLH following a measles vaccination who recovered completely after receiving immunochemotherapy with cytotoxic drugs [15]. Ikebe et al. reported HLH following influenza vaccination in a 44-year-old-man with aplastic anemia undergoing allogeneic bone marrow stem cell transplantation who totally recovered after intravenous methylprednisolone and continuous intravenous infusion of tacrolimus without any cytotoxic agents [16]. Meanwhile, BNT162b2 vaccination was found to induce robust CD4+/CD8+ T cell and cytokine responses in healthy adults [17]. Recently, there were two cases of HLH following a COVID-19 vaccination: one 43-year-old female received an inactivated SARS-CoV-2 vaccine [18] and another 68-year-old man received a ChAdOx1 nCov-19 vaccine [19]. They both recovered completely either after receiving dexamethasone or only supportive care. Vaccine-unrelated EBV-HLH may present a rapidly progressive course which could not be effectively controlled with corticosteroid use only. According to the report by Kogawa et al., most patients (60%) require a multi-agent chemotherapeutic regimen, including corticosteroids, etoposide, and cyclosporine. Especially those patients with both hyperbilirubinemia (>1.8 mg/dL) and hyperferritinemia (>20,300 ng/mL) at the time of diagnosis had significantly poorer outcomes [20]. In contrast, the clinical course of vaccine-related EBV-HLH showed less aggressive and easier to be controlled with only corticosteroids or conservative management [18,19]. Our patient presented a rapidly deteriorating unstable hemodynamic condition with both hyperbilirubinemia and hyperferritinemia initially, but the disease soon stabilized after IVIG and methylprednisolone pulse therapy given, which is unlike typical EBV-HLH in the past. The fulminant presentation of the HLH in our case could be the net result of both acute immune-stimulation after COVID-19 vaccination and EBV infection.

## 4. Conclusions

We reported a previously healthy adolescent presenting with fulminant HLH after BNT162b2 vaccination who recovered after IVIG, steroids, and ECMO support. Our case suggests that the immune activation after COVID-19 vaccination is likely to interfere with the adequate immune response to certain infectious pathogens, resulting in a hyperinflammatory syndrome. It is important for clinicians to consider HLH in patients who develop persistent fever with cytopenia following vaccination.

## Figures and Tables

**Figure 1 vaccines-10-00573-f001:**
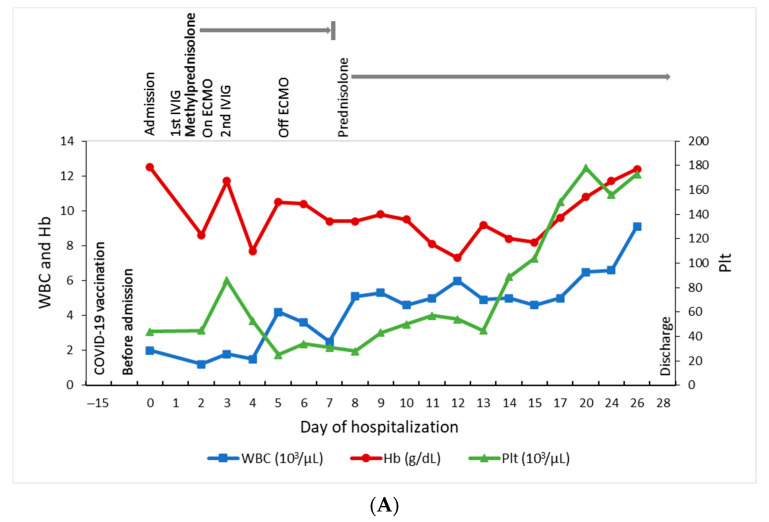
Summary of the clinical course. (**A**) Dynamic changes of the blood cell counts. (**B**) Dynamic changes of the ferritin, IL-6, and CRP levels.

**Figure 2 vaccines-10-00573-f002:**
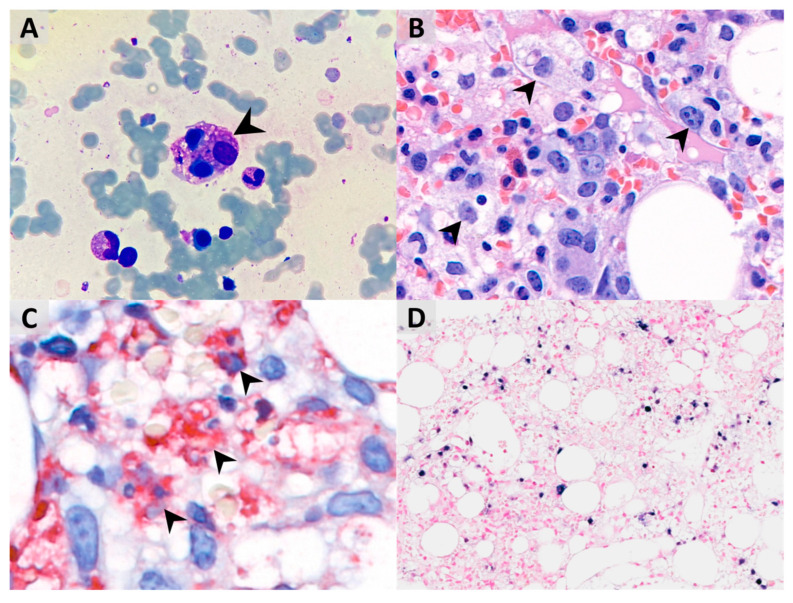
(**A**) Bone marrow aspirate smear shows hemophagocytosis of normoblasts (indicated by arrowheads, Liu’s stain, 1000× magnification). (**B**) Bone marrow biopsy shows significantly increased foamy histiocytes, which seem to phagocytize other cells (hematoxylin and eosin (H&E) stain, 800× magnification). (**C**) Hemophagocytic histiocytes are highlighted by the CD68 immunohistochemical stain with red cytoplasmic staining (1000× magnification). (**D**) Some lymphocytes are positive for the EBER in situ hybridization and show nuclear dark-purple staining (200× magnification).

## Data Availability

Not applicable.

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
