# Peer review of "Hemophagocytic Lymphohistiocytosis Following BNT162b2 mRNA COVID-19 Vaccination"

_vaccines, 2022, doi:10.3390/vaccines10040573_

Round 1

Reviewer 1 Report

Major comments:

The manuscript has clearly and concisely been written. Two questions. One is whether development of EBV-HLH coincidently occurred after BNT162b2 vaccination? Or closely related with vaccination? Since BNT162b2 vaccine was given two weeks before and that this disease was severe EBV-HLH, how to differentiate this case from vaccine-unrelated EBV-HLH? Also, although EBV VCA-IgM was negative, was this EBV-HLH not due to primary EBV infection? How about the titer of VCA-EADR-IgG? Second question is if this HLH episode occurred under hereditary HLH predisposition (FHL type 2-5). Although the patient was 14 years old, cases of late onset FHL have been reported. Ideally, flowcytometry of perforin expression in T or NK cells and CD107a expression on stimulated T or NK cells are required to rule out such possibility.

Minor comments:

  1. Line 68; Epstein-Barr-encoding-region (EBER); Commonly, EBER represents Epstein-Barr virus-encoded small RNAs.
  2. Line 124; fetal HLH; fatal HLH? or life-threatening HLH?

Reviewer 2 Report

the work is very interesting; the observations of serious adverse events associated with infection or vaccination in this pandemic contingency are very useful and also contribute to shedding light on many pathologies correlate with a cytokine storm syndromes.

Finally this case report is suitable for the Journal.

to be considered for publication, however, it needs a revision, as follow:

Line 20 and line 40:  fatal? Please explain and revise.

Line 57: please, quantify  hypofibrinogenemia

Line 86: please provide more details on EBV-DNA quantification methods and sensibility, due to the relevance it has in the clinical history.

Is the 17 days the last time point without further controls, until discharge after 28 days or thereafter?

And what about antibodies kinetic against EBV  (timepoints, titres).

Line 93: please provide details on respiratory panel: many readers may be unfamiliar with this diagnostic

Discussion: Please add a comment on EBV concomitant infection and clearance and corticosteroid therapy. Oral prednisolone therapy is not included in the figure 1b.

Overall, concomitant EBV infection is not fully discussed and EBV has not been investigated and quantified longitudinally in saliva and bone marrow.

This is a flaw in the report

Round 2

Reviewer 1 Report

Further comments:

 1.What are significant digits?

      1,980,518 IU/mL (518 necessary?)

       1151.89pg/mL (Do all numbers necessary?)

       26963.55 pg/mL (Do all numbers necessary?)

2. This is a female case, thus no need to mention X-linked lymphoproliferative disease type I and type II

3. The comment that most of cases with vaccine-unrelated EBV-HLH showed rapidly progressive course and not effective only under corticosteroid use, which require chemo-immunotherapy with the HLH-2004 protocol may be too strong and not correct. If the authors look at other papers such as Kogawa K, et al. Pediatr Blood Cancer. 2014;61(7):1257-1262, in which 60% of the patients required this protocol but the other 40% did not.

Author Response

RESPONSES TO REVIEWERS’ COMMENTS

Reviewer: 1

Point 1: What are significant digits?

1,980,518 IU/mL (518 necessary?)

1151.89pg/mL (Do all numbers necessary?)

26963.55 pg/mL (Do all numbers necessary?)

Response 1: Thank you for this comment.

Some digits are not significant. We had changed “1,980,518 IU/mL” to “1,980,000 IU/mL”, “1151.89 pg/mL” to “1151 pg/mL”, and “26963.55 pg/mL” to “26963 pg/mL”. We also amended this part in the manuscript accordingly. Please refer to line 97, 106, and 107.

Point 2: This is a female case, thus no need to mention X-linked lymphoproliferative disease type I and type II

Response 2: Thank you for this comment.

We had removed “X-linked lymphoproliferative disease type I and type II “. We also amended this part in the manuscript accordingly. Please refer to line 194.

Point 3: The comment that most of cases with vaccine-unrelated EBV-HLH showed rapidly progressive course and not effective only under corticosteroid use, which require chemo-immunotherapy with the HLH-2004 protocol may be too strong and not correct. If the authors look at other papers such as Kogawa K, et al. Pediatr Blood Cancer. 2014;61(7):1257-1262, in which 60% of the patients required this protocol but the other 40% did not.

Response 3: Thank you for this valulable comment. We agree with reviewer’s comment that this is sentence is too strong. We had ammended this part in the last paragraph as below.

Vaccine-unrelated EBV-HLH may presented a rapidly progressive course which could not be effectively controlled with corticosteroid use only. According to the report by Kogawa K, et al., most patients (60%) required a multi-agent chemotherapeutic regimen, including corticosteroid, etoposide, and cyclosporine. Epecially those patients with both hyperbilirubinemia (>1.8 mg/dl) and hyperferritinemia (>20,300 ng/ml) at the time of diagnosis had significantly poorer outcomes [20]. In contrast, the clinical course of vaccine-related EBV-HLH showed less aggressive and easier to be controlled with only corticosteroids or conservative management [18, 19]. Our patient presented a rapidly deteriorating unstable hemodynamic condition with both hyperbilirubinemia and hyperferritinemia initially, but the disease soon stabilized after IVIG and methylprednisolone pulse therapy given, which is unlike typical EBV-HLH in the past. The fulminant presentation of the HLH in our case could be the net result of both acute immune-stimulation after COVID-19 vaccination and EBV infection.

Reviewer 2 Report

the authors have appropriately modified the work in relation to the comments

Author Response

Thank you for your valuable comment.